# Ischaemic Stroke Occurring in a Patient Treated with Monoclonal Antibodies for COVID-19

**DOI:** 10.3390/v15061235

**Published:** 2023-05-25

**Authors:** Antonio Lobasso, Ciro di Gennaro, Maria Rita Poggiano, Antonio Vasta, Raffaele Angelo Nicola Ranucci, Roberto Lobianco, Anna Giacoma Tucci, Enrico Cavaglià, Pierpaolo Di Micco

**Affiliations:** 1UOC Medicina, P.O. A. Rizzoli, ASL Napoli 2 Nord, Lacco Ameno, 80076 Naples, Italy; 2AFO Medicina, P.O. Santa Maria delle Grazie, ASL Napoli 2 Nord, Pozzuoli, 80078 Naples, Italy; 3UO Radiology, P.O. A. Rizzoli, ASL Napoli 2 Nord, Lacco Ameno, 80076 Naples, Italy; 4UOC Radiology, P.O. Santa Maria delle Grazie, ASL Napoli 2 Nord, Pozzuoli, 80078 Naples, Italy

**Keywords:** COVID-19, ischaemic stroke, chronic lymphatic leukaemia, sotrovimab

## Abstract

Since the COVID-19 outbreak began, an association between COVID-19 and thrombotic diseases has been underlined. Although this association is more frequent with venous thromboembolism, ischaemic stroke has also been reported as a thrombotic complication in several cohorts of affected patients. Furthermore, the association between ischaemic stroke and COVID-19 has been considered a risk factor for early mortality. On the other hand, after the successful vaccination campaign, the incidence and the virulence of SARS-CoV-2 decreased, though it has been observed that COVID-19 may induce a severe infection in specific cohorts of frail subjects. For this reason, different drugs have been introduced of an antiviral action in order to improve the disease outcome of frail patients. In this field, with the arrival of a neutralizing monoclonal antibody against SARS-CoV-2, in particular, sotrovimab, a further chance to treat high-risk patients with mild-to-moderate COVID-19 arrived, achieving a concrete reduction in the risk of disease progression. We here report our clinical experience of an ischaemic stroke occurring a few minutes after the administration of sotrovimab for the treatment of moderate COVID-19 in a frail patient with chronic lymphocytic leukaemia. Other causes of ischaemic stroke were ruled out, and in order to evaluate the probability of a rare side effect, the Naranjo probability scale has also been utilized. In conclusion, among several side effects that have been described during the treatment of COVID-19 with sotrovimab, ischaemic stroke was not reported. Therefore, we here report a rare case of ischaemic stroke with early clinical manifestation after the administration of sotrovimab for the treatment of moderate COVID-19 in an immunocompromised patient for the first time.

## 1. Background

Since the COVID-19 pandemic due to SARS-CoV-2 began, an overlapping of thrombotic manifestations has been reported in patients with COVID-19 [1]. Although venous thromboembolism (VTE) is the more common thrombotic disorder described in COVID-19 patients, with the specific reporting of morbidity and mortality, other thrombotic diseases such as ischaemic stroke and myocardial infarction have also been described [2]. Ischaemic stroke may occur in patients with COVID-19 for several reasons; among them, the prothrombotic ability of the infection [3] and the presence of further cardiovascular risk factors [4] seem to play a relevant role.

Although the incidence of ischaemic stroke in patients with COVID-19 does not increase, the risk of pulmonary embolism or venous thrombosis not only increase but their prognosis may be severe in inpatients with COVID-19 [5].

After a long vaccination campaign, the morbidity and mortality of SARS-CoV-2 infection seems to be reduced, although it may remain with considerable virulence in inpatients with specific comorbidities that can induce frailty with a poor prognosis independent of the patient’s vaccination status [6]. For these cohorts of frail patients, besides the chance to be treated with antivirals, there is now an opportunity to be treated with monoclonal antibodies against SARS-CoV-2, such as sotrovimab, in order to improve disease outcomes. Sotrovimab is a recombinant engineered human IgG1 monoclonal antibody that binds a highly conserved epitope on the spike (S) protein receptor binding domain (RBD) of SARS-CoV-2 with a high affinity, but it does not compete with human angiotensin-converting enzyme 2 receptor binding. It acts as a monoclonal antibody, and it mimics the physiological immunological action against the spike protein of SARS-CoV-2 [7]. Moderate and severe COVID-19 with associated comorbidities may be treated with sotrovimab according to the clinical evidence of a phase 1/2/3 randomised, double-blind, placebo-controlled clinical trial conducted with a cohort of 583 non-hospitalised adults, where the use of the drug improved the outcomes of frail patients and decreased related mortality to 7% [8]. Several low-intensity side effects were reported in the trials, and they usually occurred more frequently in patients who were treated with a placebo rather than in those who were receiving sotrovimab (i.e., 2% vs. 6%, respectively); however, diarrhoea was the most common side effect reported in the group of patients treated with sotrovimab (i.e., 1%). Formerly, the list of side effects of sotrovimab reported in experimental models and in clinical trials did not include ischaemic stroke [7,8]. Because this clinical complication (i.e., ischaemic stroke) has not been described in patients treated with sotrovimab, but may have a strong impact on clinical outcome per se and in combination with COVID-19 [2,4], we here report a unique case of ischaemic stroke which occurred a few minutes after the administration of sotrovimab in a frail patient with moderate COVID-19 and decreased immunological power due to a chronic leukaemia.

## 2. Case History

C.E. A 74-year-old woman was admitted to the emergency room of the P.O. Rizzoli in Ischia Island because of the occurrence of dyspnoea and fever (i.e., 37.2 °C). Her anamnesis revealed diabetes and chronic lymphatic leukaemia in the clinical follow up. A physical examination revealed tachypnoea, and haemogas analysis revealed respiratory alkalosis with hypocapnia. Blood samples were taken, and these revealed a slight increase in C reactive protein with leucocytosis, although her white blood cell count was similar to the last value taken a few months ago. Furthermore, anaemia with haemoglobin 8.9 g\dL and moderate thrombocytopenia (i.e., platelets 65.000 mmcube) were detected. The blood tests are summarised in Table 1. Because the symptoms were dyspnoea and fever, and because lymphatic chronic leukaemia induces a decrease in immunological power, a nasopharyngeal swab to test the presence of SARS-CoV-2 was performed, and the result of this test was positive (real time PCR, STANDARD™ M10 SARS-CoV-2, Relab—molecular innovation). For this reason, a chest high-resolution CT scan was also performed that revealed COVID-19 interstitial pneumonia with a CHUNG score of 12/25 (Figure 1).

Because patients with haematological malignancy have reduced immunological power and are at risk of developing severe COVID-19, the patient was addressed to adjuvant therapy with monoclonal antibodies (i.e., sotrovimab) associated with standard treatment (e.g., ventilator support, antibiotics, steroids, and low-molecular-weight heparin).

The administration of sotrovimab started 18 h after the patient’s admission to hospital, and it was delivered according to international protocol (i.e., a single infusion of 500 mg, administration i.v. in a 100 mL sodium chloride diluted infusion in 30 min). Sotrovimab was administered with enoxaparin 4000 u daily, ceftriaxone 2 g iv, and methylprednisolone 60 mg daily. These drugs were administered according to the associated bleeding risk due, in particular, to moderate thrombocytopenia and, according to posology, was suggested by the results of glomerular filtrate and liver function tests (see Table 1).

A few minutes after the infusion of sotrovimab, the patient developed dysarthria and right hemiparesis with left facial droop. ECG was normal in sinus rhythm, blood pressure was regular (i.e., 130/70 mmHg), glycaemia was normal 90 mg/dL, and the Glasgow Coma Scale was 13. Immediately, a cerebral CT scan was performed in order to exclude intracranial haemorrhage because the patient was at increased bleeding risk for thrombocytopenia, but the result of this test was normal. The patient’s ABCD risk score was five, with moderate risk of developing ischaemic stroke, so a further cerebral CT scan was performed after 18 h, but this also did not reveal cerebral ischaemia, though neurological dysfunction was persistent. In the meantime, the patient was addressed to antiplatelet treatment because her neurological dysfunction with treated with aspirin 250 mg daily; thrombolysis was not considered because of thrombocytopenia. The patient’s NIHSS was less than 16; for this reason, a further examination in the form of cerebral MR was performed after 48 h from the onset of symptoms. The MR revealed a subtentorial ischaemic lesion of pons (Figure 2), confirming the thrombotic complication despite the bleeding risk.

Furthermore, because the neurological signs were bilateral, a further cerebral MR was performed after 72 h, and this confirmed the radiological evolution of the ischaemic lesion of pons and did not reveal further ischaemic contralateral lesions.

Most causes of ischaemic stroke are immediately evaluated in order to prevent also the risk of recurrence or other thrombotic complications. For this reason, an ECG Holter of 24 h was performed, which did not reveal arrhythmia, and a carotid ultrasound scan was also performed, and this did not reveal haemodynamic carotid stenosis. Furthermore, thrombophilia screening was performed to look for levels of protein C, protein S, AT III, homocysteinemia, anticardiolipin antibodies, lupus anticoagulans, and also to look for prothrombin A20210G and/or factor V Leiden gene polymorphism. The thrombophilia screening did not reveal any type of prothrombotic alteration (Table 1). Cerebral air embolism was excluded mainly due to personal anamnesis (i.e., the absence of recent thoracic surgery) [9]. Similarly, in order to thoroughly evaluate the chance of a rare event occurring as a side effect of sotrovimab administration, the Naranjo probability scale was used. The results were comprised between five and eight, estimating this clinical picture as a probable side effect.

After 4 days of antiplatelet treatment, the patients’ neurological symptoms improved, and they were addressed to rehab.

In the meantime, their COVID-19 improved without clinical complications or respiratory rehab. After 7 days of treatment, the patient’s NPS was negative (real time PCR, STANDARD™ M10 SARS-CoV-2, Relab—molecular innovation). Overlapping bacterial and/or fungal infections were also ruled out by serial blood culture and sputum culture during hospitalisation. Therefore, we may postulate that this is the first clinical report that has testified an ischaemic stroke after the administration of monoclonal antibodies against COVID-19.

## 3. Discussion

Risk factors for ischaemic stroke include several clinical conditions, such as age, gender, smoking, hypertension, diabetes, dyslipidaemia, familial history of atherothrombosis, obesity [10], thrombophilia [11], chronic inflammatory diseases [12], and arrhythmias with or without valves’ diseases [13]. In addition, during acute infections, a prothrombotic state may also be generated, with an increased risk of developing thrombosis, as we learnt during the COVID-19 pandemic [14]. Inpatients with COVID-19 suffered more frequently venous thromboembolism compared to ischaemic stroke, but patients that suffered ischaemic stroke had the same relevant increased mortality [5,15].

Furthermore, together with a risk stratification for ischaemic stroke, an evaluation of bleeding risk is needed in patients with neurological dysfunctions in order to decipher whether they should be addressed for thrombolysis or other treatment [16]. Thrombolysis is discouraged in patients with thrombocytopenia and increased bleeding risk; for this reason, the patient detailed in this study was addressed to antiplatelet treatment with aspirin [16].

On the other hand, thrombotic disorders are unusual during or after the administration of monoclonal antibodies for treatment of SARS-CoV-2, such as sotrovimab [8]. It may be speculated that ischaemic stroke could be associated with other risk factors for stroke, but our clinical activity excluded other common risk factors for ischaemic stroke (i.e., arrythmia, thrombophilia, cardiac valve disease). Intriguingly, the patient showed risk factors for major bleeding, such as thrombocytopenia.

Sotrovimab has a clinical indication for the treatment of moderate and severe COVID-19 after a placebo control trial involving more than 500 patients was conducted and thrombotic complications were not found in that study. Furthermore, nor were thrombotic cerebral complications or transient ischaemic attack (TIA) reported in this study. Therefore, this can be considered the first clinical report in which the administration of sotrovimab for the treatment of moderate COVID-19 had a good outcome concerning the viral infection, but where an ischaemic stroke was suffered as a clinical complication.

Due to the complexity of the pathophysiology of the reported stroke, according to TOAST classification, it may be classified as class 4 (i.e., stroke of other determined causes) [17]. The exacerbated inflammatory response with the hypersecretion of cytokines- and viral-induced coagulation disturbances, together with prolonged hypoxemia and a form of small vessel vasculitis, increase the risk of acute cerebrovascular complications in hospitalised patients with COVID-19 [18], especially if they are frail due to underlying diseases [19]. On the other hand, major adverse cardiovascular events have been reported during infusion of other monoclonal antibodies such as tocilizumab, but this has not been reported for sotrovimab [20]. In addition, independently from what occurred in this scenario, the risk of developing stroke in patients with leukaemia is not frequent, and it has been reported more frequently during treatment [21]. Therefore, although this kind of complication will remain unusual, physicians may benefit from this information in order to underline neurological diseases or a new onset of neurological symptoms during the administration of sotrovimab.

## Figures and Tables

**Figure 1 viruses-15-01235-f001:**
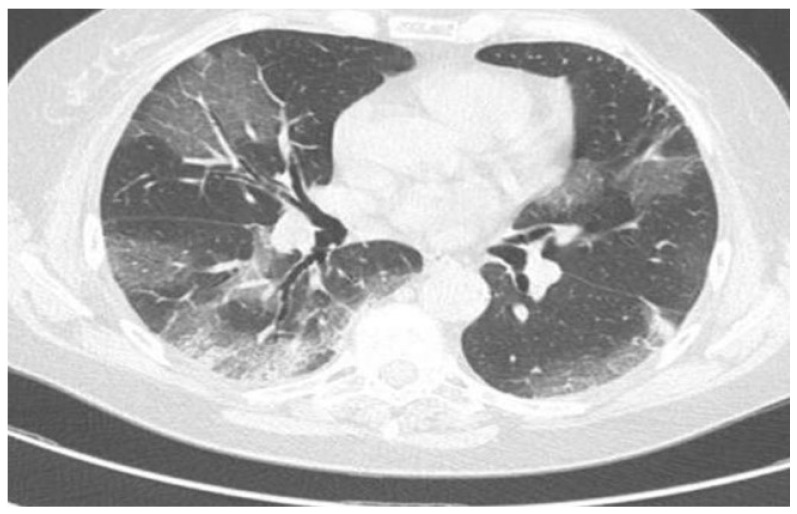
Thoracic CT scan of reported patient with moderate COVID-19.

**Figure 2 viruses-15-01235-f002:**
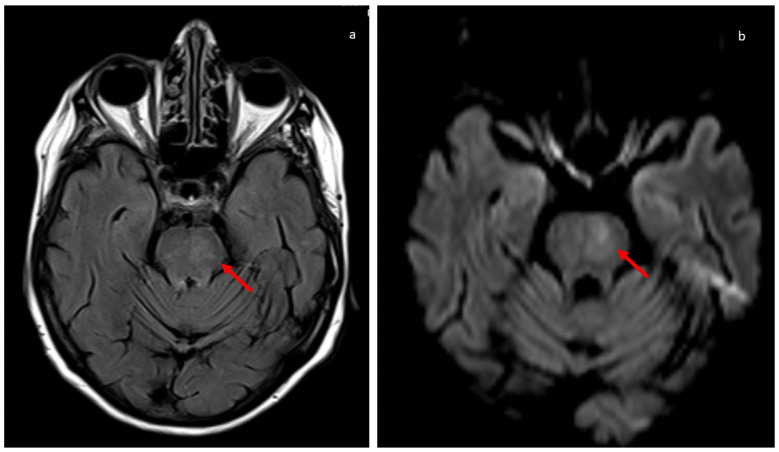
Cerebral magnetic resonance of ischaemic stroke of pons. (**a**) Axial FLAIR sequence and (**b**) axial DWI sequence show a small area of high signal intensity on the left aspect of the pons (red arrows), referable to a recent focal ischaemic lesion.

**Table 1 viruses-15-01235-t001:** Laboratory tests of reported patient.

Test	Results	Normal Values
WBC (mmcube)	65,000	4000–10,000
RBC (mmcube)	2,950,000	4,000,000–6,000,000
Hb (g\dL)	8.9	12–16
HcT (%)	27	36–48
PLT (mmcube)	65,000	140,000–400,000
PCR SARS-CoV-2	positive	Negative
Urea (mg/dL)	51	10–50
Creatinine (mg/dL)	1.1	0.5–1.1
Prothrombin time (%)	97	80–120
Activated partial thromboplastin time (s)	29	25–36
Factor V Leiden polymorphism	Wild type	Wild type
Prothrombin A20210G polymorphism	Wild type	Wild type
Protein S levels (%)	83%	50–120
Protein C levels (%)	91%	50–120
AT III levels (%)	101 (%)	60–120
d-dimer (ng/dL)	905	<500
Fibrinogen (mg/dL)	396	200–400
LAC ratio	0.96	<1.2
Anticardiolipin antibodies IgG (U/GPL)	7	<20
Anticardiolipin antibodies IgM (U/MPL)	6	<20
antiβ_2_GPI IgM isotype (U/mL)	5	<40
antiβ_2_GPI IgG isotype (U/mL)	11	<40
antiβ_2_GPI IgA isotype (U/mL)	9	<40
Homocysteine (mMol/L)	8	<15

WBC, white blood cells; RBC, red blood cells; Hb, haemoglobin; HcT, haematocrit; PLT, platelets.

## Data Availability

Not applicable.

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
