# Peer review of "Ischaemic Stroke Occurring in a Patient Treated with Monoclonal Antibodies for COVID-19"

_viruses, 2023, doi:10.3390/v15061235_

Round 1

Reviewer 1 Report

The authors have reported a case of ischemic stroke in the setting of COVID-19 pneumonia in a patient treated with Sotrovimab. While the case report is noteworthy, the association of sotrovimab with arterial infarct is rather tenuous in the setting of potential confounders. I have some concerns.

1. Abstract-adequate

2. Introduction: Adequate; but should include the already established side effects (and the rates in %) of sotrovimab.

3. Case report: 

a. The neurology of the case needs revision. Is the patient right-handed? I believe the patient was dysarthric rather than aphasic because of the side and site of the stroke. Furthermore, the patient was reported to have left hemiparesis (as opposed to right hemiparesis), while the MRI showed a left pontine lesion. The report should be edited since the pyramidal tract is yet to decussate at this level. Is there a concurrent right supratentorial infarct?

b. The rate of infusion of Sotrovimab should be reported as an additional reportage of medication/intervention dosage and route

c. The Naranjo probability scale that this side effect is due to Sotorovimab should be stated.

d. Figure 1 should include the ADC sequence to corroborate the DWI sequence image, as one expects a higher MR signal intensity in 48 hours compared to the provided image.

4. Discussion

a. This should be classified as a stroke of undetermined aetiology by TOAST criteria and the authors are advised to include this in the discussion in view of negative traditional CV risk factors and the tenuous association of stroke with Sotrovimab.

b. The authors should propose the pathophysiology of arterial in the setting of COVID, Leukemia and the role of Monoclonal antibodies.

A moderate improvement in the quality of the English language of the manuscript is recommended.

Author Response

The authors have reported a case of ischemic stroke in the setting of COVID-19 pneumonia in a patient treated with Sotrovimab. While the case report is noteworthy, the association of sotrovimab with arterial infarct is rather tenuous in the setting of potential confounders. I have some concerns.

We thank referee 1 for his/her positive comments.

  1. Abstract-adequate

We reviewed abstract according to reviewer’s suggestions

  1. Introduction: Adequate; but should include the already established side effects (and the rates in %) of sotrovimab.

We thank referee 1 for this suggestion. Rates of complications during sotrovimab administration have been added

  1. Case report: 
  2. The neurology of the case needs revision. Is the patient right-handed? I believe the patient was dysarthric rather than aphasic because of the side and site of the stroke. Furthermore, the patient was reported to have left hemiparesis (as opposed to right hemiparesis), while the MRI showed a left pontine lesion. The report should be edited since the pyramidal tract is yet to decussate at this level. Is there a concurrent right supratentorial infarct?

An additional revision of neurological examination has been performed by revision of clinical data and text was modified accordingly.

  1. The rate of infusion of Sotrovimab should be reported as an additional reportage of medication/intervention dosage and route.

We thank referee 1 for this request. Required data have been added accordingly.

  1. The Naranjo probability scale that this side effect is due to Sotorovimab should be stated.
  • We thank referee for this suggestion in order to confirm our clinical and pharmacological suspect. Accordingly we performed Naranjo probability scale and results gave probable score Probable ADR (5 to 8): The reaction followed a reasonable temporal sequence after a drug, followed a recognized response to the suspected drug, was confirmed by withdrawal but not by exposure to the drug, and could not be reasonably explained by the known characteristics of the patient’s clinical state.
  1. Figure 1 should include the ADC sequence to corroborate the DWI sequence image, as one expects a higher MR signal intensity in 48 hours compared to the provided image.

We looked for this image but did not find further pathological details.

  1. Discussion
  2. This should be classified as a stroke of undetermined aetiology by TOAST criteria and the authors are advised to include this in the discussion in view of negative traditional CV risk factors and the tenuous association of stroke with Sotrovimab.
  3. The authors should propose the pathophysiology of arterial in the setting of COVID, Leukemia and the role of Monoclonal antibodies.

Thanks a lot for this suggestion.  A possible pathophysiological explanation has been added in the text.

Comments on the Quality of English Language

A moderate improvement in the quality of the English language of the manuscript is recommended.

Reviewer 2 Report

Dear Authors

I would like to thank you for the opportunity of reviewing this interesting paper that is focused on a very remarkable and challenging topic that is a lively argument also in daily clinical practice. Although it has been more than 2 years since the first outbreak, the coronavirus disease 2019 (COVID-19) pandemic is still having a profound and devastating impact on global healthcare systems. Several complications can arise during ICU stay, from both COVID-19 extensive lung damage and extra-pulmonary involvement. The present case report discussed a unique case of ischemic stroke occurring after few

minutes after the administration of sotrovimab in a frail patient with COVID-19 and decreased immunological power due to chronic leukemia.

This paper is pleasurable to read, although it suffers from some limitations that Authors can easily adjust to improve their review making it more eligible for this important Journal. Furthermore, the Authors can improve some sections of the paper, adding information and including other important references about this topic that, in my opinion, should be cited and discussed. 

Although the language used is quite appropriate, I (I am not a native English speaker) recommend that the Authors obtain a certified native speaker with proficiencies in the scientific-medical field to complete this paper properly (if not yet done). Moreover, I recommend making a further revision of the manuscript to fix some small typing/language errors. For example, “Because patients with hematological malignancy have reduced immunological power and RE at risk to develop severe COVID-19”.

Although the introduction fits the context of the study, some concepts could be more clearly explicated in an exhaustive introduction, which would help readers to become passionate about reading the paper and using it as a reference. For example, the pathological mechanism behind neurological complications related to COVID-19 should be more deeply described. Acute cerebrovascular events such as acute ischemic stroke and intracranial hemorrhage are the most common neurological complication of COVID-19. Both these neurological complications appear to involve coagulation disturbances secondary to direct viral invasion through olfactory pathways or the bloodstream, with consequent endothelial damage. Moreover, the exacerbated inflammatory response with hypersecretion of cytokines, together with prolonged hypoxemia and a form of small vessel vasculitis, further increases the risk of acute cerebrovascular complications. This could explain why cerebrovascular complications seem to affect patients with COVID-19 that are younger and have no history of vascular abnormality compared to patients with ischemic stroke/intracranial hemorrhages due to other causes. Acute neurologic findings were recorded in 14% of COVID-19 patients on admission to the ICU and stroke was demonstrated to be more frequent in this scenario compared to hospitalized patients (5-6% vs. 1-3%) [N Engl J Med. 2020 Jun 4;382(23):2268-2270. doi: 10.1056/NEJMc2008597] [Diagnostics (Basel). 2022 Mar 29;12(4):846. doi: 10.3390/diagnostics12040846].  Please cite both the aforementioned references.

Please state if other microbiological agents (both bacterial and fungal) were investigated and how.

Furthermore, “Someone may speculate that ischaemic stroke could be associated with other risk factors for stroke but our clinical activity excluded other common risk factors for ischaemic stroke (i.e. arrythmia, thrombophilia, cardiac valve disease). Intriguingly, the patients showed risk factors for major bleedings as thrombocytopenia.” Numerous procedures can result in venous air embolism. Examples include central line placement, hemodialysis line placement, intravenous contrast injection, pacemaker/defibrillator placement, radiofrequency cardiac ablation and, rarely, even peripheral intravenous placement [Anaesth Intensive Care. 2010 Jan;38(1):175-84. doi: 10.1177/0310057X1003800127][ J Clin Med. 2016 Oct 31;5(11):93. doi: 10.3390/jcm5110093][ Diagnostics (Basel). 2017 Jan 17;7(1):5. doi: 10.3390/diagnostics7010005]. Considering the severity of the case and the invasive procedures performed, an iatrogenic air embolism cannot be excluded. Please discuss this possibility in the text.

Please also provide a figure showing COVID-19 pneumonia on the chest CT of the same patient.

Tables 1 and 2 should be merged and normal values simply be reported between brackets after the patient’s value.

Finally, I think references should be reformatted as suggested by Viruses Author’s guidelines (Author 1, A.B.; Author 2, C.D. Title of the article. Abbreviated Journal Name YearVolume, page range)

Best regards, 

Author Response

Point by point response to referee 2

Dear Authors

I would like to thank you for the opportunity of reviewing this interesting paper that is focused on a very remarkable and challenging topic that is a lively argument also in daily clinical practice. Although it has been more than 2 years since the first outbreak, the coronavirus disease 2019 (COVID-19) pandemic is still having a profound and devastating impact on global healthcare systems. Several complications can arise during ICU stay, from both COVID-19 extensive lung damage and extra-pulmonary involvement. The present case report discussed a unique case of ischemic stroke occurring after few

minutes after the administration of sotrovimab in a frail patient with COVID-19 and decreased immunological power due to chronic leukemia.

This paper is pleasurable to read, although it suffers from some limitations that Authors can easily adjust to improve their review making it more eligible for this important Journal. Furthermore, the Authors can improve some sections of the paper, adding information and including other important references about this topic that, in my opinion, should be cited and discussed. 

  • We thank referee 2 for these positive comments, we tried to modifiy text accordingly.

Although the language used is quite appropriate, I (I am not a native English speaker) recommend that the Authors obtain a certified native speaker with proficiencies in the scientific-medical field to complete this paper properly (if not yet done). Moreover, I recommend making a further revision of the manuscript to fix some small typing/language errors. For example, “Because patients with hematological malignancy have reduced immunological power and RE at risk to develop severe COVID-19”.

  • We thank referee 2 for this suggestion. A new revision of English style has been performed.

Although the introduction fits the context of the study, some concepts could be more clearly explicated in an exhaustive introduction, which would help readers to become passionate about reading the paper and using it as a reference. For example, the pathological mechanism behind neurological complications related to COVID-19 should be more deeply described.

  • We thank referee 2 for this suggestion; further info on neuropathological actions of COVID-19 have been added.

Acute cerebrovascular events such as acute ischemic stroke and intracranial hemorrhage are the most common neurological complication of COVID-19. Both these neurological complications appear to involve coagulation disturbances secondary to direct viral invasion through olfactory pathways or the bloodstream, with consequent endothelial damage. Moreover, the exacerbated inflammatory response with hypersecretion of cytokines, together with prolonged hypoxemia and a form of small vessel vasculitis, further increases the risk of acute cerebrovascular complications. This could explain why cerebrovascular complications seem to affect patients with COVID-19 that are younger and have no history of vascular abnormality compared to patients with ischemic stroke/intracranial hemorrhages due to other causes. Acute neurologic findings were recorded in 14% of COVID-19 patients on admission to the ICU and stroke was demonstrated to be more frequent in this scenario compared to hospitalized patients (5-6% vs. 1-3%) [N Engl J Med. 2020 Jun 4;382(23):2268-2270. doi: 10.1056/NEJMc2008597] [Diagnostics (Basel). 2022 Mar 29;12(4):846. doi: 10.3390/diagnostics12040846].  Please cite both the aforementioned references.

  • We thank referee 2 for this suggestion, we adequate text and references accordingly.

Please state if other microbiological agents (both bacterial and fungal) were investigated and how.

We thank referee 2 for this suggestion. Overlapping of infections were excluded and this information now appears in the text.

Furthermore, “Someone may speculate that ischaemic stroke could be associated with other risk factors for stroke but our clinical activity excluded other common risk factors for ischaemic stroke (i.e. arrythmia, thrombophilia, cardiac valve disease). Intriguingly, the patients showed risk factors for major bleedings as thrombocytopenia.” Numerous procedures can result in venous air embolism. Examples include central line placement, hemodialysis line placement, intravenous contrast injection, pacemaker/defibrillator placement, radiofrequency cardiac ablation and, rarely, even peripheral intravenous placement [Anaesth Intensive Care. 2010 Jan;38(1):175-84. doi: 10.1177/0310057X1003800127][ J Clin Med. 2016 Oct 31;5(11):93. doi: 10.3390/jcm5110093][ Diagnostics (Basel). 2017 Jan 17;7(1):5. doi: 10.3390/diagnostics7010005]. Considering the severity of the case and the invasive procedures performed, an iatrogenic air embolism cannot be excluded. Please discuss this possibility in the text.

  • We thank referee 2 for his/her suggestion. This risk factor has been mentioned and discussed in the text accordingly.

Please also provide a figure showing COVID-19 pneumonia on the chest CT of the same patient.

  • We added a related image of HRCT of chest.

Tables 1 and 2 should be merged and normal values simply be reported between brackets after the patient’s value.

  • Tables 1 and 2 have been merged accordingly.

Finally, I think references should be reformatted as suggested by Viruses Author’s guidelines (Author 1, A.B.; Author 2, C.D. Title of the article. Abbreviated Journal Name YearVolume, page range).

  • We reviewed references’ style accordingly.

Round 2

Reviewer 1 Report

Thank you for the opportunity to review this manuscript again. The current revision has improved the quality of the manuscript. I still have few comments.

1.  Did the authors mean that the ADC was normal by this statement "We looked for this image but did not find further pathological details"? An MRI done within 48 hours after a stroke is highly unlikely to reveal a normal ADC sequence. A pseudo normalization of ADC (subacute MRI findings around 1-3 weeks) is not expected at 48hr. This is more so when one considers the apparent FLAIR/DWI mismatch between Image a and b. The authors are advised to comment on this finding of "normal ADC"

This revision has lots of typographical and grammatical errors, further eroding the quality of its English language. The manuscript now needs a thorough English language revision while paying close attention to typos.

Author Response

Thank you for the opportunity to review this manuscript again. The current revision has improved the quality of the manuscript. I still have few comments.

  1. Did the authors mean that the ADC was normal by this statement "We looked for this image but did not find further pathological details"? An MRI done within 48 hours after a stroke is highly unlikely to reveal a normal ADC sequence. A pseudo normalization of ADC (subacute MRI findings around 1-3 weeks) is not expected at 48hr. This is more so when one considers the apparent FLAIR/DWI mismatch between Image a and b. The authors are advised to comment on this finding of "normal ADC"

We agree with comment of referee 1 and we are really sorry for misunderstanding.

In truth our response was based on the fact that we did not find further cerebral ischaemic lesion with the control , but of course we find the normal radiological evolution of an ischaemic lesion. In particular we did not find contralateral ischaemic lesion nor further homolateral ischaemic lesion, so confirming that the lesion of the pons was the only appearance of the ischaemic stroke and neurological symptoms were similar to a Millar-Gubler syndrome although not all ocular symptoms were present. So from a technical point of view we did not add the further image (no changes in the clinical picture nor in the radiological evolution), of course we could do it if it is fundamental for the referee.

However, this crucial clinical point is now better described in the text (see the paragraph case history)

Reviewer 2 Report

The authors addressed raised points appropriately.

Author Response

we thank referee 2 for his/her positive comment
